

# Error in jump height estimation using the flight time method: simulation of the effect of ankle position between takeoff and landing

Carlos Gonçalves[1], Roberto Baptista[2], James Tufano[3], Anthony J. Blazevich[4] and Amilton Vieira[1]

[1] University of Brasília, Faculty of Physical Education, Brasília, Distrito Federal, Brazil
[2] University of Brasília, Faculty of Technology, Brasília, Distrito Federal, Brazil
[3] Faculty of Physical Education and Sport, Charles University Prague, Prague, Czech Republic
[4] Centre for Human Performance, School of Medical and Health Sciences, Edith Cowan University, Joondalup, WA, Australia

## ABSTRACT

During vertical jump evaluations in which jump height is estimated from flight time (FT), the jumper must maintain the same body posture between vertical takeoff and landing. As maintaining identical posture is rare during takeoff and landing between different jump attempts and in different individuals, we simulated the effect of changes in ankle position from takeoff to landing in vertical jumping to determine the range of errors that might occur in real-life scenarios. Our simulations account for changes in center of mass position during takeoff and landing, changes in ankle position, different subject statures (1.44–1.98 m), and poor to above-average jump heights. Our results show that using FT to estimate jump height without controlling for ankle position (allowing dorsiflexion) during the landing phase of the vertical jump can overestimate jump height by 18% in individuals of average stature and performing an average 30 cm jump or may overestimate by ≤60% for tall individuals performing a poor 10 cm jump, which is common for individuals jumping with added load. Nevertheless, as assessing jump heights based on FT is common practice, we offer a correction equation that can be used to reduce error, improving jump height measurement validity using the FT method allowing between-subject fair comparisons.

# INTRODUCTION

Muscular fitness monitoring is common in health and sports fields, with vertical jump testing being one of the simplest, quickest, most informative, and most common tests available. Since vertical jump testing can assess the capability of the lower limbs to maximally elevate the center of mass (CoM), this test is used to quantify lower body power output, making it helpful for exercise prescription (*Read et al., 2016*), quantifying acute and long-term effects of exercise interventions (*Kibele, 1998*; *Caserotti, Aagaard & Puggaard,*

Corresponding author
Carlos Gonçalves,
carlos.goncalves@aluno.unb.br

*2008*; *Rivière et al., 2020*) , and assessing athlete's ability to return to sport after injury (*Jordan et al., 2020*; *O'Malley et al., 2018*). Vertical jump testing has been used to evaluate neuromuscular function in youth (*Fernandez-Santos et al., 2015*) and older individuals (*Singh et al., 2014*), as well as for monitoring disease and dysfunction (*Buchan et al., 2010*), people with obesity (*Bellicha et al., 2020*), and others. Given the widespread use of vertical jump testing, it is paramount to have a thorough understanding of all aspects of the test that could affect validity and reliability.

Force platforms are often used for jump testing and are considered to be a gold-standard for jump height measurement, with the takeoff velocity method (*McMahon et al., 2018*) ($h = v^2.(2.g)^{-1}$) demonstrated to produce valid and reliable data. However, force platforms require substantial investment, making alternative devices like jump mats, optical systems, and smartphone apps popular in sports performance and rehabilitation settings. Alternative devices typically estimate jump height using the flight time (FT) method ($h = g.t^2.8^{-1}$), and while a plethora of studies demonstrate high test reliability (*Bogataj et al., 2020a*; *Bogataj et al., 2020b*; *Cruvinel-Cabral et al., 2018*; *Pueo et al., 2017*), test validity has been questioned by studies reporting systematic jump height overestimation (*Kibele, 1998*; *García-López et al., 2005*; *Moir, 2008*; *Wade, Lichtwark & Farris, 2019*).

Regardless of the well-known overestimation issue, practitioners often accept the error, assuming that the error is relatively constant among trials (*i.e.,* the same amount of error would exist for the same athlete maintaining the same technique over time). Nevertheless, the FT method requires the precise determination of the instant of takeoff and landing, requiring high sampling rates of devices (>100 Hz). In this regard, the advances in smartphone technology have allowed jump recording at 240 Hz, making jump measurement more accessible and precise. However, the FT method requires jumpers to takeoff and land with the same body posture, which is technically challenging and makes precise analyses difficult. Even after providing a habituation (familiarization) session, researchers have observed that lower limb joints are more flexed (ankle more dorsiflexed) at landing than at takeoff (*Wade, Lichtwark & Farris, 2019*). While the tester can request the jumper repeat the test, this is only possible when the tester can detect the change in body posture in real time. An experienced assessor may easily detect large changes in hip and knee joint angles, but smaller changes (~10°) at the ankle joint may be very hard to detect yet significantly affect FT. Previous studies (*Yamashita, Murata & Inaba, 2020*; *Shu et al., 2016*; *Wade, Lichtwark & Farris, 2019*) have consistently reported changes between 10 and 20 degrees in ankle position between takeoff and landing, and the ankle joint has been pointed out as the principal contributor to errors in jump height estimation (*Yamashita, Murata & Inaba, 2020*).

Important to the above arguments, many studies that have shown acceptable FT jump estimates when conducted over a short period, typically a few days to two weeks (*Bogataj et al., 2020a*; *Bogataj et al., 2020b*; *Cruvinel-Cabral et al., 2018*; *Pueo et al., 2017*; *García-López et al., 2005*; *Gallardo-Fuentes et al., 2016*). It could be speculated that the jumper has a greater chance of changing their body posture over longer periods since jump testing is used mainly by researchers in longitudinal studies to monitor training adaptations. Accounting for changes in body posture during jumping maybe even more critical when

clinicians use jump test results as the criterion for athletes to return to competition since changes in movement patterns would be expected as athletes recover from injury (*Jordan et al., 2020*). Therefore, the present study investigated the effect of changes in ankle position from takeoff to landing in vertical jumping. To provide a robust number of possibilities, our simulations accounted for a range of human statures (from 1.44 to 1.98 m) and individuals performing "below average", "average", and "above average" jumps (*McKay et al., 2017*). In addition, we proposed a correction equation to account for individuals who might not follow the proper landing technique (*e.g.,* people with cognitive impairment), whereby the jump data would need to be corrected. For practical reasons, this study investigated only changes at the ankle joint since the ankle seems to be the joint most poorly inspected during jump testing. However, it is essential to note that changes at other joints (*e.g.,* knee, hip, or torso) will affect jump height estimates.

## MATERIALS & METHODS

### Biomechanical model

A four-segment model (torso, thigh, shank, and feet) (*Winter, 2009*) was created to evaluate the vertical CoM distance from the floor (Fig. 1). Additionally, the tiptoe-to-ankle (lateral malleolus) horizontal distance was set at 78.7% of the foot length (*Tilley & Henry Dreyfuss Associates, 2001*) (Fig. 2). The motion of the model was constrained to the sagittal plane while it stood on a single leg with doubled expected mass. The kinematic chain origin was set at the tiptoes, and a set of 2D rotational matrices (*Robertson et al., 2013*) was used to create the expected motion of the ankle, knee, and hip joints. The kinematic chain rotations provided the CoM position for each body segment (foot, shank, hip, and torso). The whole-body CoM was calculated from a weighted sum of the segment's CoM position and mass, with CoM vertical displacement exclusively resulting from changes in ankle position. Therefore, the maximum change in body posture between jump takeoff and landing was set at landing with feet flat (angle joint at 0°).

### Simulated jumps and research variables

We used a continuum of 100 body heights ranging from 1.435 m (3 standard deviations (SD) below the women's average) to 1.984 m (3 SD above the men's average) (*WHO, 2007*) , which was required to estimate foot length (*Winter, 2009*) (15.2% of the stature) and subsequently ankle position. The simulations also accounted for jump heights of 0.10, 0.20, 0.30, and 0.40 m allowing the simulation of 16,400 jumps (100 stature × 41 angle positions × 4 jump heights). The takeoff velocity and ascending time were calculated using the constant acceleration equations. The error in jump height estimation was computed for each condition. The difference in CoM position due to a change in ankle position was defined as any difference between landing and takeoff (nomenclature definitions at Table 1):

$$h_{diff} = h_l - h_{to}. \tag{1}$$

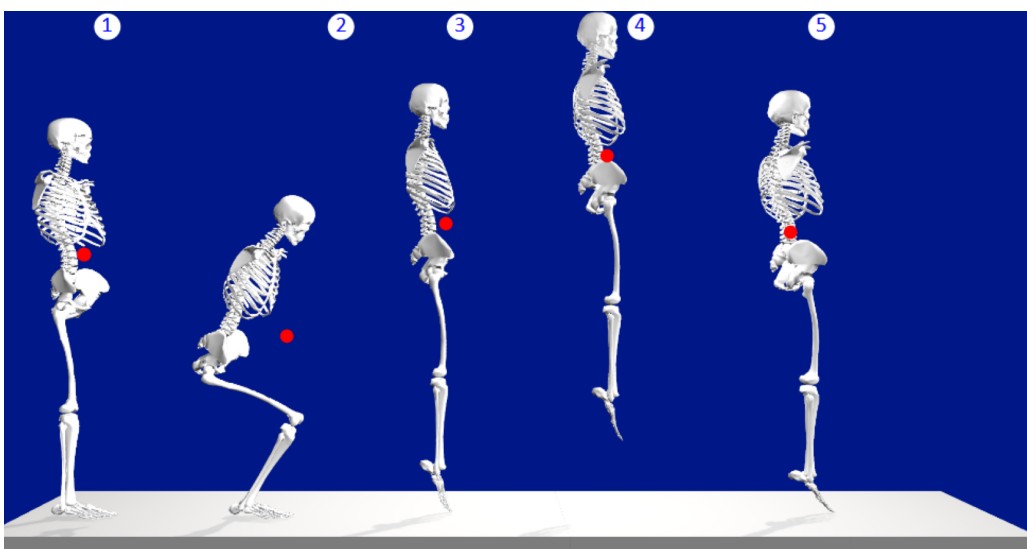

**Figure 1** **Center of mass displacement during jump.** A typical vertical jump of a scaled model of 1.75 m in stature with a 20° change in ankle position from takeoff (3) and landing (5). The vertical CoM distances from the floor (1 = 1.16 m, 2 = 0.89 m, 3 = 1.29 m, 4 = 1.59, and 5 = 1.27 m) were represented. Image source credit: Images from OpenSim software.

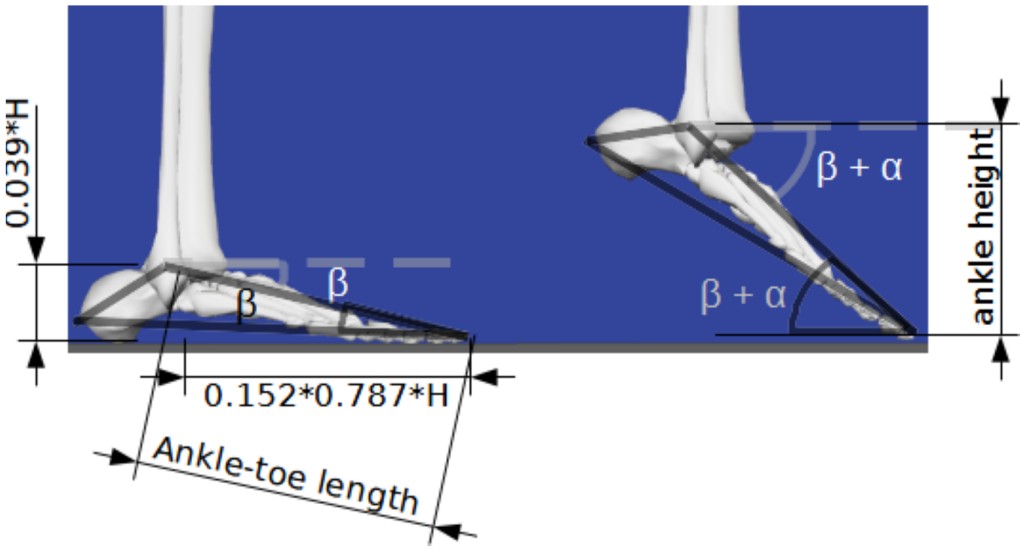

**Figure 2** **Ankle height estimation with anthropometric estimates.** Anthropometric estimates derived from individual stature (H) and ankle position during jump takeoff and landing. $\beta$ and $\alpha$ angles were used to calculate ankle height. $\beta$ represent the natural foot angle relative to the ground during standing posture, while $\alpha$ represents the plantarflexion performed during jump takeoff and landing. Image source credit: Images from OpenSim software.

**Table 1 Nomenclature.**

| | | | |
|---|---|---|---|
| $g$ | Gravitational acceleration ($ms^{-2}$) | $FT_{true}$ | True flight time (s) |
| H | Stature (m) | $V_{to}$ | Vertical velocity at takeoff (s) |
| $h$ | Simulated jump height (m) | $FT$ | Measured flight time (s) |
| $h_{to}$ | Center of Mass height at takeoff (m) | $FT_{diff}$ | Flight time difference (s) |
| $h_l$ | Center of Mass height at landing (m) | $\hat{h}$ | Estimated jump height (m) |
| $h_{diff}$ | Center of Mass height difference (m) | $\hat{h}_c$ | Corrected estimated jump height (m) |
| $t_{diff}$ | Time difference due to change in body posture | $h_{error}$ | Percentage error in jump height estimation (%) |
| $\beta$ | Natural foot angle relative to the ground during standing posture | $t_a$ | Ascending time (s) |
| $\alpha$ | Ankle position performed during jump | | |

The ascending time ($t_a$) is directly related to jump height (h), the takeoff velocity ($V_{to}$), and the true flight time ($FT_{true}$), *i.e.*, the CoM height is the same at the takeoff and landing of the jump. Therefore,

$$t_a = \sqrt{\frac{2.h}{g}}, \tag{2}$$

$$FT_{true} = 2.t_a, \tag{3}$$

$$V_{to} = g.t_a. \tag{4}$$

Consequently, the measured FT when there was any change in ankle position was estimated as:

$$FT = \frac{V_{to} + \sqrt{V_{to}^2 - 2.g.h_{diff}}}{g}. \tag{5}$$

The percentage difference in jump height estimations was defined as $h_{error}$, where $\hat{h}$ is the approximation using the measured FT.

$$\hat{h} = \frac{FT^2.g}{8}, \tag{6}$$

$$h_{error} = \left(\frac{\hat{h} - h}{h}\right).100. \tag{7}$$

When there is any change in ankle position (or any change in body posture) between takeoff and landing, the presented equations can be manipulated to obtain $FT_{true}$ from $FT$ and $t_{diff}$ (time difference due to change in body posture), as follows:

$$FT_{true} = FT - t_{diff}, \tag{8}$$

$$t_a = \frac{1}{g} \cdot \left( \frac{g.FT}{2} - \frac{h_{diff}}{FT} \right), \tag{9}$$

$$h = \frac{1}{2.g} \cdot \left( \frac{g.FT}{2} - \frac{h_{diff}}{FT} \right)^2. \tag{10}$$

Obtaining $h_{diff}$ is challenging without a motion capture system. Therefore, we evaluated the potential of estimating $h_{diff}$ ($\hat{h}_{diff}$) based on ankle-toe length ($l_{at}$) estimated from stature (H), and ankle position: ankle angle at takeoff ($\alpha_{to}$), and landing ($\alpha_{land}$), acquired from 2D video analysis (Fig. 2).

$$l_{at} = \sqrt{(0.039H)^2 + (0.152 \times 0.787H)^2} = 0.126H, \tag{11}$$

$$\hat{h}_{diff} = l_{at} \left( \sin(\alpha_{to} + \beta) - \sin(\alpha_{land} + \beta) \right), \tag{12}$$

therefore, the "corrected" jump height ($\hat{h}_c$) can be estimated as:

$$\hat{h}_c = \frac{1}{2.g} \cdot \left( \frac{g.FT}{2} - \frac{\hat{h}_{diff}}{FT} \right)^2. \tag{13}$$

## RESULTS

Our results show that not controlling the ankle position (*i.e.*, allowing dorsiflexion) during the landing phase of the vertical jump can overestimate jump height up to 59.6% using the FT method (Fig. 3). Tall individuals jumping around 0.10 m were those most affected by changes in ankle position. Still, individuals of average stature (1.71 m) performing an average jump of 0.30 m may have their jump height estimated with an error of 18.4%.

We applied Eq. (13) to correct the simulated jump heights more prone to greater error, *i.e.*, ≤0.20 m. As a result, we obtained a perfect relationship between $\hat{h}_c$ and $h$ ($F(1, 8198) = 7e + 08$, $p < 0.001$, $r^2 = 1.00$).

## DISCUSSION

We investigated the effect of changing ankle position from takeoff to landing during vertical jumping on jump height estimations using the FT method. We considered an extensive range of statures, small to large changes in ankle position, and "poor" to "above average" jump height performances.

We demonstrated that jump height might be overestimated by 59.6% for a tall person (1.98 m) landing with the ankle completely flat (real jump height of 10 cm, while the

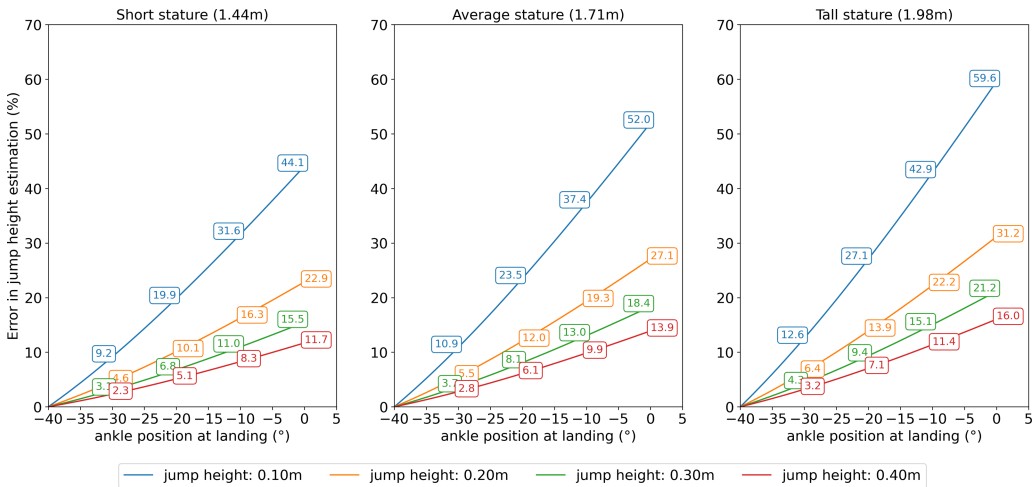

**Figure 3** **Results for error in jump height estimation.** Error in jump height estimation using flight time method. Measurement error accounting for individual's stature, changes in ankle position, and jump height.

estimated was 15.96 cm) using the FT method. However, even a person with an average stature (1.71 m) with an average jump performance of 30 cm can present a significant error in their jump height estimation of 8.1 or 13.0%, changing the ankle position in 20° or 30°, respectively. Yet smaller changes from 10° to 20°, as average changes previously reported (*Yamashita, Murata & Inaba, 2020*; *Shu et al., 2016*; *Wade, Lichtwark & Farris, 2019*), can significantly (∼6.5%) affect jump height accuracy making the measurement error probably greater than the observed improvement in jump height (∼2.1%) following weeks of exercise training (*Lindberg et al., 2023*). We subsequently propose a "correction equation" to deal with individuals who might inadvertently change ankle position during vertical jump testing since small changes in ankle position (∼10°) will probably be present (*Wade, Lichtwark & Farris, 2019*) but may not the detectable to the human eye.

Our results corroborate previous experimental data (*Kibele, 1998*; *Moir, 2008*; *Aragón, 2000*) reporting that FT method tends to overestimate jump height (up to ∼4 cm) compared to criterion measures (*e.g.,* impulse-momentum method), suggesting that participants landed with some part of their bodies partially flexed. Changes in any body joint between takeoff to landing can turn CoM closer to the floor and artificially increase the flight time. A study (*Yamashita, Murata & Inaba, 2020*) with participants performing countermovement jumps with arm swing demonstrated that the changes in foot position were the major source of error (2.5 of 5.3 cm change in CoM). Surprisingly, this error was greater than the error caused by changes in arm position (1.7 cm), which is another well-known source of error in jump height estimation using the FT method. Other studies reported differences of ∼10° in ankle position from jump takeoff to landing, which might be similar (*Kibele, 1998*) or even smaller (*Wade, Lichtwark & Farris, 2019*) than changes in knee position. However, changes in knee position are probably easier to observe in real-time than changes at the ankle joint.

The current study demonstrated that individuals with greater foot length performing a "below average" jump are more prone to their jump height being estimated poorly. Therefore, it could be speculated that taller individuals are more inclined to have their jump overestimated since they usually have longer feet and possibly greater body mass, which in turn makes the jump performance more challenging, contributing to poor jump height. In addition, lower jump height would be expected for those performing overloaded jump testing, which has become popular in testing batteries for force profiling with participants jumping as low as 10 cm (*Samozino et al., 2013*). On the other hand, as jump height was increased in the models used in the present study, the error in jump height estimation decreased (Fig. 3).

We have also demonstrated that Eq. (13) corrects the poorer estimated jump heights, making measured jump heights similar to their true values. It is important to note that Eq. (13) requires only an estimate of the change in ankle position (*e.g.,* 10°), stature or foot length, or ideally, ankle-toe distance. Therefore, this procedure could be helpful for those using smartphone applications (*Vieira et al., 2023*) or any device using the FT method to estimate jump height. In addition, researchers or practitioners may use Eqs. (9) and (10) to evaluate the jump height error where the difference in CoM position between the takeoff and landing is measurable since flexion at any joint (torso, hip, and knee) on landing would decrease the distance from vertical CoM position and the foot contact area to the floor. It should be noted that the benefit of jump height correction must be greater than the cost of implementing a joint angle measurement. We might assume that researchers requiring greater internal validity can find that benefit overcomes this cost. We also judge that the proposed correction should be applied to make comparisons between subjects fair.

It is important to note that we focused on variations in ankle position (*i.e.,* ankle flexion angle) and did not simulate all sources of accuracy disturbances using the FT method. Besides the effect of changes in body posture between takeoff and landing, the time measurement resolution is an additional source of error. These error components should be simulated in future research proposing average confidence intervals for jump height estimates that use the FT method, especially in smaller jumps performed by taller individuals. Additionally, our simulations were generated with individual statures, and caution should be taken using the results of a per-user evaluation. Direct measurements should give more accurate results.

Complementary, we know that estimating jump height from flight time (or any other outcome metric from biological systems) will inherently be affected by natural biological variability or error from any source (*e.g.,* equipment). While variability or error cannot be entirely eradicated, testing familiarization can improve accuracy. In this study, we are alerting for the proper landing technique and our results should encourage future data acquisition from individuals with diverse anthropometrics and performance characteristics. It also needs to determine whether jump familiarization or verbal command (*e.g.,* "land on the balls of your feet") might affect the accuracy of jump height estimations.

**Auxiliary tool**

Using this calculator https://www.lptf.unb.br/calculadora the estimated jump can be automatically corrected by inserting (1) a change in ankle position and (2) stature or foot length or, ideally, ankle-toe distance. For changes in ankle position at landing, we have provided information for setting up the recording device (*e.g.,* smartphone) and video processing using the freely available Kinovea software (https://www.kinovea.org/). The dataset used for this research is also available for consulting.

## CONCLUSIONS

Changes in ankle position from takeoff to landing in vertical jumping can overestimate jump height by up to 60% using the flight time method. Tall individuals with low jump heights are likely more prone to larger errors. However, even individuals of average stature performing an average jump height may have their jump poorly estimated, with a 18% error, according to the present data. On the other hand, using the simple calculator provided in this study can reduce the error and improve jump height validity.

### Funding

The APC was funded by Edital DPI/DPG/BCE 01/2024 from the University of Brasilia. The funder had no role in study design, data collection and analysis, decision to publish, or preparation of the manuscript.

### Grant Disclosures

The following grant information was disclosed by the authors:
Edital DPI/DPG/BCE 01/2024 from the University of Brasilia.

### Competing Interests

The authors declare there are no competing interests.

### Author Contributions

- Carlos Gonçalves conceived and designed the experiments, performed the experiments, analyzed the data, prepared figures and/or tables, authored or reviewed drafts of the article, and approved the final draft.
- Roberto Baptista analyzed the data, authored or reviewed drafts of the article, and approved the final draft.
- James Tufano analyzed the data, authored or reviewed drafts of the article, and approved the final draft.
- Anthony J. Blazevich analyzed the data, authored or reviewed drafts of the article, and approved the final draft.
- Amilton Vieira conceived and designed the experiments, analyzed the data, prepared figures and/or tables, authored or reviewed drafts of the article, and approved the final draft.

## Data Availability

The raw data and Jupyter scripts are available in the Supplemental Files.

## Supplemental Information

Supplemental information for this article can be found online at http://dx.doi.org/10.7717/peerj.17704#supplemental-information.

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
