# Peer review of "Error in jump height estimation using the flight time method: simulation of the effect of ankle position between takeoff and landing"

_PeerJ, doi:10.7717/peerj.17704_

## Round 0.1 · original submission · Minor Revisions

Dear authors, congratulations on your work. Reviewers have made important comments including the need to reinforce the argument regarding the value of the changes to be expected and improve the discussion in light of practical applications.

Please, consider the reviewer's suggestions a re-submmit a new version for further evaluation.

Regards

Reviewer 1 ·

Basic reporting

The authors investigated the error in jump height estimation using the flight time method, considering the changes in ankle knee position between takeoff and landing. This is a simulation based on an anthropometric model.
I found the study very interesting and brings important information to the literature.

Experimental design

The study appears to have no important issues from a methodological point of view.

Validity of the findings

I have just a few comments to discuss and question the practical applications of the study.

It is recognized that all motor tests used in physical assessment inherently entail biological error and variability, which cannot be entirely eradicated. What should be prioritized is a thorough familiarization of individuals to mitigate such errors. With this in mind, the authors should contemplate for a future study conducting an error analysis in a real-world setting with well familiarized subjects, calculating the variability (error), and subsequently determining whether it falls within acceptable margins.

Another aspect to consider is that in an evaluation, the paramount concern is the consistency of measurements. Even if the estimated metric doesn't precisely correspond to the actual metric (i.e. agreement), what matters most is its reliability. This is crucial when monitoring the acute or chronic effects of training.

In regards to proposing an equation to adjust the values, I found the spreadsheet quite interesting. However, what concerns me is the feasibility of acquiring angular measurements, which could potentially pose a significant challenge in field assessments.

Please try to contemplate the aspects mentioned above in your discussion.

Reviewer 2 ·

Basic reporting

The study is relatively well written and clear. The information provided is sufficient for the reader to understand the context. For further clarity, please consider the use of single 'terms' throughout, e.g. ankle joint angle. There are appear to be some errors with referencing (maybe some field code error) and some references appear complete.

Experimental design

There was sufficient detail & information to replicate they study, supported by appropriate references.

Validity of the findings

The findings from the simulated data showed large errors. The authors discuss their findings in the context of athletic training, rehabilitation etc. This context, however, is not substantiated. For example, an example of flat-footed landing is presented (very unlikely to happen, realistically, and easily detectable by the tester) as well as an example of change of 10 degrees (30 degrees - 20 degrees). None of these examples, however, appear to be based on literature. It would help the argument if some indication of experimental data could provide /support the value of the changes to be expected (e.g. https://www.ncbi.nlm.nih.gov/pmc/articles/PMC5120166/) to strengthen the argument.

Additional comments

The calculator is very useful, based on the what the study is discussion. The manuscript also suggests that information was provided on setting up the recording device and the subsequent analysis. I was not able to find those. In any case, if those refer to the foot image (Fig 2), then a screenshot from these measurements from the Kinovea window or an annotated image would have been clearer.

---

## Round 0.2 · accepted · Accept

Dear authors,
The reviewers considered that all issues pointed out were adequately addressed and recommended the acceptance for publication. Congratulations on your work.

Reviewer 1 ·

Basic reporting

The authors have successfully addressed all aspects of my previous revision, so I recommend accepting the paper.

Experimental design

No comments

Validity of the findings

No comments

Additional comments

No comments

Reviewer 2 ·

Basic reporting

I am satisfied with all the amendments / responses to my previous comments.

Experimental design

I am satisfied with all the amendments / responses to my previous comments.

Validity of the findings

I am satisfied with all the amendments / responses to my previous comments.

Additional comments

I am satisfied with all the amendments / responses to my previous comments.